# Determinants of the food insecurity at household level in Pakistan: A multilevel model approach

Tahir Mahmood[1], Ramesh Kumar[2,3]*, Tariq Mehmood Ali[2], Nawal Naeem[2], Sathirakorn Pongpanich[3]

1 Faculty of Business Administration, Sukkur IBA University, Sukkur, Pakistan, 2 Department of Public Health, Health Services Academy, Islamabad, Pakistan, 3 College of Public Health Sciences, Chulalongkorn University, Bangkok, Thailand

* drramesh1978@gmail.com

**Data Availability Statement:** All relevant data are within the paper and its Supporting Information files.

## Abstract

Food insecurity is a major concern for the developing world and around 37% of the population of Pakistan is food insecure. This paper utilizes the Food Insecurity Experience Scale (FIES) to assess the population prevalence of food insecurity and to identify their risk factors and determinants at the household level in Pakistan. This study employs a multi-level random coefficient model, using the Pakistan Panel Household Survey (PPHS-2010) dataset; representative data from 4,130 households. Factors like; income of the household, gender, education, household size, land ownership, and shocks of food insecurity allow the incidence of idiosyncratic shocks (injuries and/or casualties) at the community level, which affects the food insecurity situation of the community, rather differently were included. The study confirms a statistically significant inverse relationship between household income, household size, and household head education with food insecurity and a positive association of shocks and inflation with food insecurity at the household level. Specifically, with increasing per capita income of the household, food insecurity declines (coefficient: -0.083, statistically significant at 1%) and food insecurity increases with shocks (coefficient: 0.058, statistically significant at 1% significance level). The study also reveals a significant heterogeneity at a one percent significance level in the determinants of food insecurity at the district, community, and household levels. The income of the household, household head gender and education level, household size, household assets, shocks, injuries, and inflationary pressure are important determinants of food insecurity in Pakistan.

## Introduction

Globally, food insecurity has become a significant public health issue, with a marked increase in the last decade [1, 2]. One in three persons is affected by malnutrition, undernutrition, micronutrient deficiencies, and being underweight in the world [1–3]. Using estimates and comparing the determinants at national, regional, and global levels, conventional model-based

**Funding:** The author(s) received no specific funding for this work.

**Competing interests:** The authors have declared that no competing interests exist.

estimates such as; calorie consumption, household food expenditures, individual dietary intake, and anthropometry can provide significant results for food insecurity [4–7]. Food insecurity is a multi-faceted issue that includes regional, clusters, and characteristics of the household [8, 9]. The Food Insecurity Experience Scale (FIES) method is used to measure food insecurity at the household level in Pakistan [10]. Ending hunger and malnutrition is the most daunting global problem as 828 million populations are facing food insecurity in 2022 [11].

Around 37% of the population is facing the problem of food insecurity and 13% of the households are food insecure in Pakistan [12, 13]. During the last three decades, food insecurity is increasing at an alarming rate in developing countries [14, 15]. Pakistan is facing a rising trend in food insecurity due to political instability, economic crisis, shocks, and floods. Pakistan is ranked eleventh worst nation among 118 countries on Global Hunger Index and numerous indicators that encompass different sectors of the economy result in food insecurity [15, 16].

Previous works of literature have extensively used experiential food insecurity experience scale modules over the past decades [17–24]. The study found mixed results with per capita production of cereal, per hectare yield, an aggregate governance index, logistics performance, and employed workers being the determinants of national food security [25]. Another study conducted for 134 countries, used FIES where household food insecurity was highly associated with low education, poor social networks, low social capital, lower income, and unemployment [24]. Low levels of education, limited social capital, and living in a country with low per capita income were associated with the most severe food insecurity per FIES scores [26]. Moreover, using Gallup World Poll data, the FIES scores for elderly populations predicted the determinants such as economic and demographic determinants and several composite indices [27] and sociodemographic correlates of food insecurity among MENA economies [28] and researcher has used the conventional measures of food insecurity [29–33]. The study examined the determinants of food security in the hinterlands of Pakistan by using the multivariate logistic model and using dichotomous food security variables [29]. Another study explored the determinants of food security rather than food insecurity, in a policy framework context using data from mountainous regions of Pakistan [30]. The study used calorie intake as a measure of food insecurity using binary logistic models for estimating factors affecting household food insecurity [32].

Food insecurity is an important issue faced by an individual at the household level. Sustainable Development Goal 2 aims to achieve food security, improve nutrition and promote sustainable agriculture [11]. Pakistan is suffering from an acute shortage of food due to recent climate change threats [12]. Nevertheless, the researchers in the Pakistani context have not validated FIES. Thus, the current study investigates the determinants of food insecurity using a sophisticated methodological framework for Pakistan by using FAO, FIES [10, 17] which is based on item response theory and consists of 8 questions related to household food insecurity over a period of twelve months preceding the survey. The results of our study are comparable across Pakistan and within provinces. We have used multilevel random effect generalized linear models and incorporated numerous factors to narrow down the research gap and to guide the policymakers regarding the food insecurity situation at the micro (household) level.

## Method and materials

### Population, study design, period and study area

This study uses the Pakistan Panel Household Survey (PPHS)-2010, a third-round dataset collected by the World Bank along with the Pakistan Institute of Development Economics (PIDE) in 2010 with a representative sample from the country including all four provinces. The

PPHS-2010 was carried out at a time when inflation was escalating and the country had also suffered from some natural disasters including droughts and floods. The modules on shocks, food insecurity, subjective well-being, and overall physical security were incorporated in the third round of PPHS-2010. Therefore, the nature of the dataset is cross-sectional.

The total sample size comprises 4,142 households including both urban and rural domains. The dataset was collected for 16 districts in all the provinces of Pakistan. For the rural sample, a village or dehat is considered the PSU. The total number of rural PSUs stands out at 141. All the urban localities are divided into enumeration blocks, consisting of 200 to 250 households in each block. In total, 75 urban enumeration blocks (PSUs) were selected randomly. The final number of households selected for the study after cleaning and accounting for missing values is 4,130 with 2,790 from rural and 1,340 from urban areas. The dataset consists of indicators; literacy, employment level, poverty profile, natural shocks, nutrition, housing status, and well-being was used for food insecurity. The third round of PPHS includes the Food Insecurity Experience Scale (FIES) module, consisting of eight questions designed to evaluate the adequacy of household food requirements. These questions are related to households' behaviors and experiences regarding the difficulty in meeting their basic food need spanning over the last 12 months (see Appendix A in S1 Data for the complete questions module).

## Ethics statement

This study used PPHS survey data. Pakistan Institute of Development Economics Review Board (IRB) approved the data collection process, which involved taking written informed consent from respondents.

## Variables

The outcome variables were constructed from the FIES food security module, and the head of the household was the respondent for this study. Almost 59 percent of the households are worried about their food shortage, and 53 percent eat the same food. Moreover, at least one adult member of the household eats less than the required size of the food (44%), members of the household who skip food (21%) and do not have anything to eat the whole day (12%) for not having enough money to buy food (see Appendix A in S1 Data). The FIES items compose a statistical scale designed to cover a range of severity of food insecurity and are analyzed together as a scale. Thus, we have developed a new food insecurity matrix ranging from food-secure households to severe food-insecure households. The households who answered all 8 questions with an affirmative response (*Yes*) are placed as food-insecure households. The household that answered all 8 questions (*No*) is placed food-secure household. Further, households are classified into four categories based on experiencing different nature of the food (in) security severity such as; *Food Secure Household* (scale-0), *Mild Food Insecure Household* (scale 1–3), *Moderate Food Insecure Household* (scale 4–6) and *Severe Food Insecure Household* (7–9), [24, 34]. The main outcome variables are binary which measures the household's severity of food insecurity. The first measure, *Food Insecurity*, is equal to one if the household experienced moderate food insecurity as well as severe food insecurity within the last twelve months; zero otherwise. The other measure, *Severe Food Insecurity*, captures households experiencing the most severe range of food insecurity, and is coded as one if the household experienced severe food insecurity within the last twelve months; zero otherwise. Severe food insecurity is usually related to households experiencing physiological hunger [22]. It is justified to use two binary measures such as food insecurity and severe food-insecure households, to check the impact of the determinants of food insecurity.

The explanatory variables consist of individuals, households, and community characteristics. The severity of experiencing inflation is lowering the purchasing power of the household, which in turn affects the household food security situation. Households experiencing shocks in the form of flood, and experiencing injuries/casualties at the village level are also important determinants of food insecurity. Income is another important variable, which is used as a continuous variable by the household per capita consumption. The per capita income is an important indicator of poverty, which indicates the lesser the household income; the more food insecure and vulnerable the household is. Other important factors that impact household food insecurity are; agriculture land holding, ownership, education of the head, and household size. The education level of the household head and household size (members of a family) are used as continuous variables that affect the food insecurity of the household due to the high dependency ratio in developing countries. The gender of the head of the household is used as a determinant of food insecurity (see Appendix B in S1 Data *for* detail). It is difficult for women-headed households to work for longer hours. Therefore, it is hypothesized that women-headed households are more likely to experience food insecurity relative to male-headed households. Importantly, to avoid the loss of important information, the variables with missing values of less than 2 percent are imputed through *Multivariate Imputation by Chain Equation*.

## Data analysis

The empirical analysis utilizes *multi-level models* with random coefficients to investigate the determinants of food insecurity in Pakistan. The analysis employs a random coefficient model's specification to avoid bias from clustering households within community-level and district-level time-invariant omitted variables. Ignoring the importance of cluster effect at the community level might invalidate many of the statistical techniques. For instance, the independently and identically distributed assumptions of the regression might be violated. Therefore, multilevel models are used as a standard framework to model the statistical analysis [35]. It seems very likely that some districts and communities within districts could have been more affected by food insecurity than others, and it is very likely that the idiosyncratic and covariate shocks (i.e., inflation, flood, and injuries/casualties) that have occurred during the last twelve months would have affected the communities by different severity owing to the huge distance among the districts and peculiar geostrategic location. Consequently, we expect variation both at the district and community levels. Therefore, the theoretical underpinning of the three-level generalized linear random slope model is; the household is nested in the community (enumeration block/primary sample unit) and the communities are nested in each district (**Fig 1**).

| Multilevel analysis | Level-1: Districts |
| --- | --- |
| | → 16 districts from four provinces with different socio-economic and geographic characteristics |
| | Level-2: Communities (PSU/ Enumeration Blocks) |
| | → 215 communities with varing number of households affected by changing degree of covariate shocks (e.g., Flood, inflation) |
| | Level-3: Households |
| | → 4130 households with different socio-economic characteristics affected by changing degree of idisyncratic shocks (e.g., injuries/casualty) |

**Fig 1. Three-level multilevel hierarchical model (Random coefficient).**

The three-level multilevel random intercept model is given by Eq 1 below;

$$FI_{ijk} = \beta_0 + X_{ijk} + \gamma_k + \mu_{jk} + \varepsilon_{ijk} \tag{1}$$

Where $FI_{ijk}$ is the observed food insecurity experienced by a household "i" (i = 1, 2,. . . .4130) in community "j" (j = 1, 2, . . .. . . 215) and district "k" (k = 1, 2, . . .. 16).

Where $\beta_0$ is the mean household food insecurity at the district level, $X_{ijk}$ is the determinants of food insecurity, $\gamma_k$ is the district effect, $\mu_{jk}$ is the community effect and $\varepsilon_{ijk}$ is the household level residual error term.

As the communities experienced 22% of the injuries/casualties, hence it is expected that the injuries have varying associations across the distribution of communities, therefore it is inevitable to allow the casualties to vary at the community level instead of keeping them constant, we use the random coefficient model. The three-level random slope generalized linear model is given by;

$$FI_{ijk} = \beta_0 + X_{ijk} + \gamma_{0k} + \gamma_{ck}X_{cijk} + \mu_{0jk} + \mu_{cjk}X_{cijk} + \varepsilon_{ijk} \tag{2}$$

Now $\gamma_{ck}X_{cijk}$ and $\mu_{cjk}X_{cijk}$ have been added as the random part of the model and additional '0' subscripts have been added to $\gamma_k$ and $\mu_{jk}$ so that they now read $\gamma_{0k}$ and $\mu_{0jk}$, respectively. Whereas, $\mu_{cjk}X_{cijk}$ random coefficient or slope model specification and $\gamma_{ck}X_{cijk}$ is the effect of district adjusted for the random explanatory variable that is community injuries/ casualty, and the subscript "$c$" represents communities or clusters.

For robustness, we tested whether the Probit Model and two-level fixed effect models are a better fit to the data set available than the three-level random coefficient model. We found that the LR test justified the use of the multilevel model and the three-level random coefficient model consistently performed better than the other models' specifications.

## Results

(Table 1) presents the socio-demographic characteristics of the respondents. The generated four categorical variables are; food-secure households (34%), mild-food-insecure households (22%), moderate food-insecure households (26%), and severe food-insecure households (17%). In the two binary generated measures; the Food Insecurity household stands at 43% while the Severe Food Insecurity Household stands at 17%. Almost 55% of the household's heads have no education and their age was 48 years on average. The household size equivalent scale is constructed through the modified Organization for Economic Co-operation and Development (OECD) equivalence scale.

(Table 2) presents the determinants of food insecurity at the household level in Pakistan. The coefficients indicate the predicted change in the outcome variable for a one-unit change in the explanatory variable, keeping everything else constant. For instance, the coefficient of gender for food-insecure households is 0.076, and for severely food-insecure households is 0.057 which shows that being female compared to male as head of the household will increase the chance of food insecurity by 0.076 units and 0.057 units, respectively. The findings confirm a significant variation of food insecurity situations across districts, communities, and at the household level at a 1% significance level. Thus, the three-level model is preferred to the single-level model. We can conclude that the 16 districts of four provinces of Pakistan, do not act as 16 independent observations; rather households were clustered by communities and districts. Similarly, the Likelihood Ratio (LR) test which compares the three-level model to the simpler two-level and single-level models, confirms that both the district level and communities' level standard deviation is independent and significant. This means households living in the same community are more the coefficient of injuries to show a significant adverse effect on

**Table 1. Individual and socio-economic characteristics of the households.**

| Variables | Mean (%) | Standard Deviation | Minimum | Maximum |
|---|---|---|---|---|
| **Outcome variables: Four categories:** | | | | |
| Food Secure Household | 0.345 (34%) | 0.475 | 0 | 1 |
| Mild Food Insecure Household | 0.220 (22%) | 0.414 | 0 | 1 |
| Moderate Food Insecure Household | 0.264 (26%) | 0.440 | 0 | 1 |
| Severe Food Insecure Household | 0.171 (17%) | 0.376 | 0 | 1 |
| **Binary categories:** | | | | |
| Food Insecure Household | 0.435 | 0.495 | 0 | 1 |
| Severe Food Insecure Household | 0.171 | 0.376 | 0 | 1 |
| **Explanatory Variables:** | | | | |
| Gender | 0.042 | 0.201 | 0 = male | 1 = female |
| Age | 47.821 | 14.585 | 15 | 90 |
| Education of Household head | 4.036 | 4.649 | 0 | 18 |
| Per capita consumption | 7.781 | 0.670 | 3.729 | 12.114 |
| Household Size | 7.602 | 3.964 | 2 | 43 |
| Employed | 0.831 | 0.374 | 0 = No | 1 = Yes |
| Own residence | 0.776 | 0.416 | 0 = No | 1 = Yes |
| Agriculture land houlding | 0.387 | 0.487 | 0 = No | 1 = Yes |
| Sanitation facilities | 0.581 | 0.493 | 0 = No | 1 = Yes |
| Shocks (floods) | 0.852 | 0.355 | 0 = No | 1 = Yes |
| No inflation | 0.096 | 0.295 | 0 | 1 |
| Mild inflation | 0.243 | 0.429 | 0 | 1 |
| Moderate inflation | 0.038 | 0.192 | 0 | 1 |
| Hihgly inflation | 0.570 | 0.495 | 0 | 1 |
| Severe inflation | 0.050 | 0.218 | 0 | 1 |
| Community wise casualties | 0.227 | 0.419 | 0 = No | 1 = Yes |
| Punjab | 0.447 | 0.497 | 0 | 1 |
| Sind | 0.294 | 0.455 | 0 | 1 |
| KP | 0.146 | 0.353 | 0 | 1 |
| Balochistan | 0.111 | 0.314 | 0 | 1 |
| Region | 0.316 | 0.465 | 0 = Rural | 1 = Urban |

household food insecurity. The coefficients of injury in the community by keeping them at random are 0.098 and 0.085 (Table 2) for insecure food households and severe food-insecure households, respectively. The results are statistically significant. Therefore, the random slope model is preferred over the fixed effect model.

## Discussion

This study found that 34% of households were food secure and 43% of households were food insecure. An Ethiopian study shows, 43% were food-secure, while the remaining 57% were food insecure. This number is increased than our estimates due to multiple factors [36]. On the contrary, another study shows that two-thirds of households were food insecure due to severe drought and lack of proper agriculture facilities [37]. The average household size in this study was 7.6 people, which is slightly more than the average household size of 6.41 [14] and the national average household size of 6.38 members was observed for the period. Almost 96% of the household head is male. Similarly, 39% of the households have agricultural landholding. Besides, 77% of the households have their own residence and 58% have access to sanitation

**Table 2. Determinants of household food insecurity: Three level multi-level models.**

| Variables | Model-1[a] | Model-2[b] |
|---|---|---|
| Gender of the household head (Reference Category male) | | |
| Female | 0.076** (0.032) | 0.057** (0.026) |
| Age of the household head as continuous variable | | |
| Age | -0.004 (0.002) | -0.001 (0.002) |
| Age Square | 0.0001 (0.0001) | 0.000 (0.000) |
| Size of the household as continuous variable | | |
| Household Size | -0.012*** (0.001) | -0.007*** (0.001) |
| Household head Education as continuous variable | | |
| Education | -0.009*** (0.001) | -0.004*** (0.001) |
| Household consumption as continuous variables | | |
| Household Consumption | -0.083*** (0.011) | -0.057*** (0.009) |
| Employment (Reference Category No) | | |
| Yes | -0.032* (0.017) | -0.028* (0.016) |
| Own residence (Reference Category No) | | |
| Yes | -0.031* (0.016) | -0.012 (0.013) |
| Agriculture land ownership (Reference Category No) | | |
| Yes | -0.030*(0.016) | -0.034** (0.013) |
| Sanitation amenities (Reference Category No) | | |
| Yes | -0.152*** (0.016) | -0.103*** (0.013) |
| Experience shocks during the last five years (Reference Category No) | | |
| Yes | 0.058*** (0.024) | -0.042** (0.019) |
| Experience inflation (Reference Category Normal inflation) | | |
| Severe inflation | 0.189 (0.048) | -0.015 (0.039) |
| interaction | 0.044 (0.050) | 0.119*** (0.041) |
| Experience injury in the community (Reference Category No) | | |
| Yes | 0.011 (0.019) | 0.019 (0.022) |
| Regional domain (Reference Category rural) | | |
| Urban | -0.021(0.025) | -0.0009 (0.015) |
| Constant | 1.269*** (0.121) | 0.749*** (0.096) |
| **Variance at district, community, household and injuries at community level** | | |
| Variance (district) | 0.166*** (0.032) | 0.121*** (0.022) |
| Variance (community) | 0.128*** (0.010) | 0.075 *** (0.008) |
| Variance (household) | 0.396***(0.004) | 0.321*** (0.003) |
| Comminuty injury | 0.098*** (0.024) | 0.085*** (0.018) |
| **Diagnostic test** | | |
| Wald chi2 (17) | 575.42 | 342.51 |
| Prob > chi2 | 0.000 | 0.000 |
| LR test vs. probit model: chi2 (3) | 351.04*** | 464.84*** |

a. Food Insecure Household Estimation

b. Severe Food Insecure Household Estimation

Robust standard errors in parentheses

* p<0.1

** p<0.05

*** p<0.01

facilities. A majority, 85% of the households have experienced covariate shocks (e.g., flood and inflation), and on average, 22% of the household experienced idiosyncratic shocks (e.g., injuries/casualties) during the last 12 months. Almost 31% of households reside in urban areas. Moreover, only 4% of females are heads of the household.

This study shows that the female compared to the male, as head of the household will increase the chance of food insecurity. This important finding is in line with previously available studies [23, 38, 39]. A study also shows that the sex of the household is a major factor that contributes to livelihood [40]. The main reason is that women have limited access to agriculture activities and livestock ownership as well as less utilization of their land [41]. Moreover, females lack opportunities in education and employment compared to their male counterparts. In contrast, one of the studies using the calorie intake method shows that female-headed households are less likely to be food insecure, it claims that being female as a household head enhances better utilization of resources, and spending more on food and health care compared to men [42].

Similarly, the current study establishes an inverse relationship between household size and food insecurity. In contrast to the conventional model estimates, the chances of being in a food-insecure household escalate with each additional member of the household [39]. Generally, the larger the number of adults in a household, the higher the chance of the household being food insecure [4, 23]. It is also plausible that larger households have higher calorie intake and thus higher food insecurity as larger households have more mouths to feed [39].

Years of schooling are an important indicator of human capital. Our study is in line with previous studies and confirms that education level is one of the important determinants of household food security [4, 23]. Moreover, the chance of experiencing food insecurity plummets with increasing years of schooling. Because, education opens up a window of new opportunities, leads to poverty reduction, enhances employment opportunities, creates self-confidence, and helps in adopting new technologies, affecting the family structure and overall living standard and welfare of the society. Through another channel, a year of schooling and food insecurity establishes an inverse relationship, which is the judicious utilization of household-limited resources.

Per-capita income is used as an important indicator of the living standard and food security of the household [42]. The Higher the income of the household, the higher will be the health status, nutrition fulfillment, and overall welfare of the household [4]. The study also confirms a significant negative relationship between per capita income and household food insecurity. Also, for an employed individual, household food insecurity declines significantly for the severe food-insecure household. Moreover, the household possessing agricultural land, and having better sanitation facilities, households' food security situation improves, significantly.

The effects of shocks such as droughts, floods, and earthquakes are quite often in a resource-constraint setting. This study confirms a significant and positive relationship between shock experience and food insecurity in both food-insecure households and severe food-insecurity households. Similarly, an important study showed that the indirect effect of drought on households' well-being is quite higher compared to the direct effect of drought on production [43]. Moreover, the immediate impact of drought on crops and livestock is quite severe. Thus, this analysis incorporates the impact of idiosyncratic shocks in the form of inflation (general rise in the price level) on food insecurity. The findings established a worsening effect on the food security of households with an increasing intensity of the inflationary shocks. Consequently, the inflationary shocks deteriorate the purchasing power, leaving households in the abysmal food insecurity trap. Importantly, an interaction term is used between the household experiencing shocks and subsequent inflationary pressure on the household. It means that households that experience shocks in the form of floods, and

drought, those households are more likely to experience inflationary pressure. Thus, the household that experience shock also experience soaring general price levels, and their combined effect on household food insecurity is alarming mainly in the severe food-insecure households. This study varied in some determinants by presenting contrasting results compared to conventional measures of food insecurity [29, 44]. Specifically, with an additional member of the household, the likelihood of food insecurity decreases and those households whose head is female, that household is more likely to be food insecure. Furthermore, the results of the multi-level null model and multi-level random effect model with only demographic and with variables that impact the food security of the household are presented in Appendix C in S1 Data.

## Conclusion

This study concludes that 34% of the households were food secure and 43% of the households were food insecure in Pakistan. Factors like; income of the household, household head gender and education level, household size, household assets, shocks, injuries, and inflationary pressure are important determinants of food insecurity. Importantly households that experience shocks in the form of flood or drought are also severely affected by soaring inflation. Their combined effect (interaction term) has a severe adverse effect on household food insecurity.

## Implications of the study for practice and policy

From a policy perspective, Pakistan needs to address food insecurity in both urban as well as rural areas through multi-sectoral intervention programs. There is an urgent need to empower and upgrade agriculture and livestock through the use of technology to facilitate the adaption of climate-smart agriculture. The study suggests either income support or cash-for-work programs are inevitable for risky food-insecure households. Moreover, it is urgent to focus on the agriculture sector specifically on small land farmers to boost their productive potential and proper land reforms. Importantly, educating farmers and giving them a sense of ownership play an important role in this regard. It is high time to introduce land reforms to increase productivity and yield per hectare.

## Limitations

We used PPHS data that lacks some important information include; agriculture data, market access, weather conditions, rainfall, drought condition, temperature, and precipitation could have a positive impact on food security. Hence, we recommend studying these important variables in future research.

## Supporting information

**S1 Data.**
(DOCX)

**S1 File.**
(DTA)

**S2 File.**
(DO)

## Acknowledgments

We acknowledge the support from the Rachadapisek Sompote Fund for Postdoctoral Fellowship, Chulalongkorn University Thailand, and support for the doctoral research from the University of Gottingen Germany.

## Author Contributions

**Conceptualization:** Tahir Mahmood.

**Data curation:** Tahir Mahmood.

**Funding acquisition:** Tariq Mehmood Ali.

**Investigation:** Tariq Mehmood Ali.

**Methodology:** Tariq Mehmood Ali.

**Project administration:** Ramesh Kumar.

**Resources:** Nawal Naeem.

**Software:** Tahir Mahmood, Nawal Naeem.

**Supervision:** Sathirakorn Pongpanich.

**Validation:** Nawal Naeem.

**Visualization:** Sathirakorn Pongpanich.

**Writing – original draft:** Ramesh Kumar.

**Writing – review & editing:** Ramesh Kumar, Nawal Naeem, Sathirakorn Pongpanich.

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
