## [Decision Letter · Decision Letter 0]

21 Jun 2023

PONE-D-23-12777Determinants of the Food Insecurity at Household level in Pakistan: A Multilevel Model Approach.PLOS ONE Dear Dr. Kumar, Thank you for submitting your manuscript to PLOS ONE. After careful consideration, we feel that it has merit but does not fully meet PLOS ONE’s publication criteria as it currently stands. Therefore, we invite you to submit a revised version of the manuscript that addresses the points raised during the review process.

 Please submit your revised manuscript by Aug 05 2023 11:59PM. If you will need more time than this to complete your revisions, please reply to this message or contact the journal office at plosone@plos.org. Please include the following items when submitting your revised manuscript:A rebuttal letter that responds to each point raised by the academic editor and reviewer(s). You should upload this letter as a separate file labeled 'Response to Reviewers'.A marked-up copy of your manuscript that highlights changes made to the original version. You should upload this as a separate file labeled 'Revised Manuscript with Track Changes'.An unmarked version of your revised paper without tracked changes. You should upload this as a separate file labeled 'Manuscript'.If applicable, we recommend that you deposit your laboratory protocols in protocols.io to enhance the reproducibility of your results. Protocols.io assigns your protocol its own identifier (DOI) so that it can be cited independently in the future. For instructions see: https://journals.plos.org/plosone/s/submission-guidelines#loc-laboratory-protocols. Additionally, PLOS ONE offers an option for publishing peer-reviewed Lab Protocol articles, which describe protocols hosted on protocols.io. Read more information on sharing protocols at https://plos.org/protocols?utm_medium=editorial-email&utm_source=authorletters&utm_campaign=protocols.

We look forward to receiving your revised manuscript.

Kind regards,

Olutosin Ademola Otekunrin

Academic Editor

PLOS ONE

Journal Requirements:

"We Acknowledge the support from Rachadapisek Sompote Fund for Postdoctoral Fellowship, Chulalongkorn University Thailand, and support for doctoral research university of Gottingen Germany"

5. Please ensure that you refer to Figure 1 in your text as, if accepted, production will need this reference to link the reader to the figure.

**Additional Editor Comments:**

Kindly, revise your manuscript by attending to the comments/suggestions of the two reviewers attached to this email.

Reviewers' comments:

Reviewer's Responses to Questions

**Comments to the Author**

1. Is the manuscript technically sound, and do the data support the conclusions?

Reviewer #1: Yes

Reviewer #2: Yes

2. Has the statistical analysis been performed appropriately and rigorously? 

Reviewer #1: Yes

Reviewer #2: Yes

3. Have the authors made all data underlying the findings in their manuscript fully available?

Reviewer #1: Yes

Reviewer #2: Yes

4. Is the manuscript presented in an intelligible fashion and written in standard English?

Reviewer #1: Yes

Reviewer #2: No

5. Review Comments to the Author

Reviewer #1: Overview: This manuscript discusses the determinants of the food insecurity at household level in Pakistan through multilevel model approach. Research methods include a questionnaire survey, and variance analysis. Overall, there are several issues with the manuscript, and I suggest that the authors thoroughly revise the manuscript based on the following comments.

For the further modification check the detailed comments attached in system!

Reviewer #2: Dear Editor

Thank you for the opportunity to review the manuscript - Determinants of the Food Insecurity at Household level in Pakistan: A Multilevel Model Approach.

General comments:

•The paper is well written, even though the authors have used old data, the food insecurity situation and determinants may have changed a lot to this point. Yet I imagine it is beneficial to understand the historical food insecurity trends to inform interventions and policy making.

•Language editing is recommended as some areas are bit difficult to read.

•I recommend the manuscript for publication, subsequent to attending to minor corrections

Abstract

Line 5 – are the determinants and risks for food insecurity at a national or household level?

Please rephrase line 13-14 it reads a bit difficult

Introduction

Please elaborate more on the food insecurity situation in Pakistan using recent sources

Add citation for line 63 to 66

Please present the background a bit more orderly – the points are a bit haphazard, maybe paragraphs can each address a theme e.g. the nature of food insecurity, SGDs and food insecurity, measurement of food insecurity, global and national food insecurity trends etc.

There is repetition on line 29-30 and 63-64

Kindly show who will benefit from this work

Methodology

Line 91 – kindly clarify, is it the head of the household who asked or responded to the questions?

Line 203 – less/fess utilization?

The stated percentages in lines 175-190 are not evident in Table 1

Results and discussion

Please discuss further using recent studies and clearly show comparisons of food insecurity and determinants in other countries

Conclusions

Kindly include the prevalence of food insecurity in the conclusion, as it has been stated in the background that the prevalence will be assessed

6. PLOS authors have the option to publish the peer review history of their article (what does this mean?). If published, this will include your full peer review and any attached files.

Reviewer #1: No

Reviewer #2: No

---

## [Author Response · Author response to Decision Letter 0]

3 Jul 2023

Dear Editor,

We have revised our manuscript in the light of comments received from the reviewers. Revised paper with response to reviewer letter along with track change file is attached for further process of publication. 

Regards

Dr Ramesh

---

## [Decision Letter · Decision Letter 1]

17 Jul 2023

PONE-D-23-12777R1

Determinants of the Food Insecurity at Household level in Pakistan: A Multilevel Model Approach

PLOS ONE

Dear Dr. Kumar,

Thank you for submitting your manuscript to PLOS ONE. After careful consideration, we feel that it has merit but does not fully meet PLOS ONE's publication criteria as it currently stands. Therefore, we invite you to submit a revised version of the manuscript that addresses the points raised during the review process.

ACADEMIC EDITOR: Kindly attend to the comments of Reviewer 2.

We look forward to receiving your revised manuscript.

Kind regards,

Olutosin Ademola Otekunrin

Academic Editor

PLOS ONE

Journal Requirements:

Reviewers' comments:

Reviewer's Responses to Questions

**Comments to the Author**

1. If the authors have adequately addressed your comments raised in a previous round of review and you feel that this manuscript is now acceptable for publication, you may indicate that here to bypass the “Comments to the Author” section, enter your conflict of interest statement in the “Confidential to Editor” section, and submit your "Accept" recommendation.

Reviewer #1: All comments have been addressed

Reviewer #2: (No Response)

2. Is the manuscript technically sound, and do the data support the conclusions?

Reviewer #1: Yes

Reviewer #2: Yes

3. Has the statistical analysis been performed appropriately and rigorously? 

Reviewer #1: Yes

Reviewer #2: Yes

4. Have the authors made all data underlying the findings in their manuscript fully available?

Reviewer #1: Yes

Reviewer #2: Yes

5. Is the manuscript presented in an intelligible fashion and written in standard English?

Reviewer #1: Yes

Reviewer #2: No

6. Review Comments to the Author

Reviewer #1: I appreciate the author for the incorporation of all the comments and the revised version.

Good Luck

Reviewer #2: Thank you to the authors for attending to the comments. However the following comment was not sufficiently addressed.

Line 99 - Kindly clarify why the questions were asked by the head of the household? it also contradicts with the statement that follows. It reads as if the head of the household asked and responded to the questions, which does not sound right.

7. PLOS authors have the option to publish the peer review history of their article (what does this mean?). If published, this will include your full peer review and any attached files.

Reviewer #1: No

Reviewer #2: No

---

## [Author Response · Author response to Decision Letter 1]

25 Jul 2023

Dear Editor,

A revised manuscript is attached for further revision and publication in your esteemed journal.

Regards

Dr Ramesh

---

## [Decision Letter · Decision Letter 2]

9 Aug 2023

PONE-D-23-12777R2Determinants of the Food Insecurity at Household level in Pakistan: A Multilevel Model ApproachPLOS ONE

Dear Dr. Kumar,

Thank you for submitting your manuscript to PLOS ONE. After careful consideration, we feel that it has merit but does not fully meet PLOS ONE’s publication criteria as it currently stands. Therefore, we invite you to submit a revised version of the manuscript that addresses the points raised during the review process. Please, kindly attend to the minor comments of Reviewer 2.Please submit your revised manuscript by Sep 23 2023 11:59PM. If you will need more time than this to complete your revisions, please reply to this message or contact the journal office at plosone@plos.org. Please include the following items when submitting your revised manuscript:A rebuttal letter that responds to each point raised by the academic editor and reviewer(s). You should upload this letter as a separate file labeled 'Response to Reviewers'.A marked-up copy of your manuscript that highlights changes made to the original version. You should upload this as a separate file labeled 'Revised Manuscript with Track Changes'.An unmarked version of your revised paper without tracked changes. You should upload this as a separate file labeled 'Manuscript'.If applicable, we recommend that you deposit your laboratory protocols in protocols.io to enhance the reproducibility of your results. Protocols.io assigns your protocol its own identifier (DOI) so that it can be cited independently in the future. For instructions see: https://journals.plos.org/plosone/s/submission-guidelines#loc-laboratory-protocols. Additionally, PLOS ONE offers an option for publishing peer-reviewed Lab Protocol articles, which describe protocols hosted on protocols.io. Read more information on sharing protocols at https://plos.org/protocols?utm_medium=editorial-email&utm_source=authorletters&utm_campaign=protocols.

We look forward to receiving your revised manuscript.

Kind regards,

Olutosin Ademola Otekunrin

Academic Editor

PLOS ONE

Journal Requirements:

Reviewers' comments:

Reviewer's Responses to Questions

**Comments to the Author**

1. If the authors have adequately addressed your comments raised in a previous round of review and you feel that this manuscript is now acceptable for publication, you may indicate that here to bypass the “Comments to the Author” section, enter your conflict of interest statement in the “Confidential to Editor” section, and submit your "Accept" recommendation.

Reviewer #1: All comments have been addressed

Reviewer #2: (No Response)

2. Is the manuscript technically sound, and do the data support the conclusions?

Reviewer #1: Yes

Reviewer #2: Yes

3. Has the statistical analysis been performed appropriately and rigorously? 

Reviewer #1: (No Response)

Reviewer #2: Yes

4. Have the authors made all data underlying the findings in their manuscript fully available?

Reviewer #1: Yes

Reviewer #2: Yes

5. Is the manuscript presented in an intelligible fashion and written in standard English?

Reviewer #1: Yes

Reviewer #2: No

6. Review Comments to the Author

Reviewer #1: I have no comment for the author. I highly appreciate the authors commitment and their understanding of the Issues.

Good luck!

Reviewer #2: I suggest that line 98 be rephrased as follows: The outcome variables were constructed from the FIES food security module, and the head of the household was the respondent for this study.

7. PLOS authors have the option to publish the peer review history of their article (what does this mean?). If published, this will include your full peer review and any attached files.

Reviewer #1: No

Reviewer #2: No

---

## [Author Response · Author response to Decision Letter 2]

12 Aug 2023

Reviewer Response letter is attached.

---

## [Decision Letter · Decision Letter 3]

29 Aug 2023

Determinants of the Food Insecurity at Household level in Pakistan: A Multilevel Model Approach PONE-D-23-12777R3Dear Dr. Kumar,We pleased to inform you that your manuscript has been judged scientifically suitable for publication and will be formally accepted for publication once it meets all outstanding technical requirements. Within one week, you'll receive an e-mail detailing the required amendments. When these have been addressed, you'll receive a formal acceptance letter and your manuscript will be scheduled for publication. An invoice for payment will follow shortly after the formal acceptance. To ensure an efficient process, please log into Editorial Manager at http://www.editorialmanager.com/pone/, click the Update My Information link at the top of the page, and double check that your user information is up-to-date. If you have any billing related questions, please contact our Author Billing department directly at authorbilling@plos.org.

Kind regards,

Olutosin Ademola Otekunrin

Academic Editor

PLOS ONE

Additional Editor Comments (optional):

Reviewers' comments:

Reviewer's Responses to Questions

**Comments to the Author**

1. If the authors have adequately addressed your comments raised in a previous round of review and you feel that this manuscript is now acceptable for publication, you may indicate that here to bypass the “Comments to the Author” section, enter your conflict of interest statement in the “Confidential to Editor” section, and submit your "Accept" recommendation.

Reviewer #1: All comments have been addressed

Reviewer #2: All comments have been addressed

2. Is the manuscript technically sound, and do the data support the conclusions?

Reviewer #1: Yes

Reviewer #2: Yes

3. Has the statistical analysis been performed appropriately and rigorously? 

Reviewer #1: Yes

Reviewer #2: Yes

4. Have the authors made all data underlying the findings in their manuscript fully available?

Reviewer #1: Yes

Reviewer #2: Yes

5. Is the manuscript presented in an intelligible fashion and written in standard English?

Reviewer #1: Yes

Reviewer #2: Yes

6. Review Comments to the Author

Reviewer #1: Author made a great modification in revised version manuscript and now as to me it's enough for publication requirements

.

Reviewer #2: Please move line 99-103 ''Almost 59 percent .... (see Appendix A).'' to the results section, since these are part of the results.

7. PLOS authors have the option to publish the peer review history of their article (what does this mean?). If published, this will include your full peer review and any attached files.

Reviewer #1: No

Reviewer #2: No

---

## [Editor Report · Acceptance letter]

27 Sep 2023

PONE-D-23-12777R3 

Determinants of the Food Insecurity at Household level in Pakistan: A Multilevel Model Approach 

Dear Dr. Kumar:

I'm pleased to inform you that your manuscript has been deemed suitable for publication in PLOS ONE. Congratulations! Your manuscript is now with our production department. 

Kind regards, 

on behalf of

Dr. Olutosin Ademola Otekunrin 

Academic Editor

PLOS ONE